# The Therapeutic Role of Plastic and Reconstructive Surgery in the Interdisciplinary Treatment of Soft-Tissue Sarcomas in Germany—Cross-Sectional Results of a Prospective Nationwide Observational Study (PROSa)

**DOI:** 10.3390/cancers14174312

**Published:** 2022-09-02

**Authors:** Benjamin Thomas, Amir K. Bigdeli, Steffen Nolte, Emre Gazyakan, Leila Harhaus, Oliver Bischel, Burkhard Lehner, Gerlinde Egerer, Gunhild Mechtersheimer, Peter Hohenberger, Raymund E. Horch, Dimosthenis Andreou, Jochen Schmitt, Markus K. Schuler, Martin Eichler, Ulrich Kneser

**Affiliations:** 1Department of Hand, Plastic and Reconstructive Surgery, Burn Center, BG Trauma Center Ludwigshafen, University of Heidelberg, 67071 Ludwigshafen, Germany; 2Department of Otorhinolaryngology, Head and Neck Surgery, Armed Forces Hospital Ulm, 89081 Ulm, Germany; 3Department of Trauma and Orthopedics, BG Trauma Center Ludwigshafen, 67071 Ludwigshafen, Germany; 4Department of Orthopaedics, Trauma Surgery and Paraplegiology, Heidelberg University Hospital, 69118 Heidelberg, Germany; 5Department of Hematology, Oncology and Rheumatology, University Hospital Heidelberg, 69120 Heidelberg, Germany; 6Institute of Pathology, Heidelberg University Hospital, 69120 Heidelberg, Germany; 7Division of Surgical Oncology, Department of Surgery, Mannheim University Medical Center, University of Heidelberg, 68167 Mannheim, Germany; 8Department of Plastic and Hand Surgery, Comprehensive Cancer Center, University Hospital of Erlangen, Friedrich-Alexander-University Erlangen-Nürnberg, 91054 Erlangen, Germany; 9Department of General Orthopedics and Tumor Orthopedics, University Hospital Münster, 48149 Münster, Germany; 10Department of Orthopedics and Trauma, Medical University of Graz, 8036 Graz, Austria; 11National Center for Tumor Diseases (NCT/UCC), 01307 Dresden, Germany; 12Center for Evidence-based Healthcare, University Hospital Carl Gustav Carus, Technical University Dresden, 01307 Dresden, Germany; 13Clinic and Polyclinic for Internal Medicine I, University Hospital Carl Gustav Carus, Technical University of Dresden, 01307 Dresden, Germany

**Keywords:** sarcoma, defect, reconstruction, microsurgery, reconstructive surgery, plastic surgery

## Abstract

**Simple Summary:**

The mainstay of soft-tissue-sarcoma treatment remains ablative surgery with complete tumor resection. In this context, reconstructive plastic surgery has become an important aspect of multidisciplinary sarcoma therapy aiming at limb preservation as an alternative to amputations. In this present study, cross-sectional data collected prospectively at 39 study centers across Germany were analyzed, focusing on both the inhouse availability of plastic surgery and external accessibility to plastic surgery in 621 cases. In summary, unplanned and incomplete primary tumor resections carried out at centers with lower degrees of specialization were associated with a significantly increased need for subsequent flap-based defect coverage. In line with this, a readily available team of plastic surgeons was independently associated with successful defect reconstruction, which in turn was associated with significantly higher chances of limb preservation. We conclude that easily accessible plastic surgery and a high degree of expertise in the field of sarcoma treatment are indispensable for limb preservation following sarcoma resection. Plastic and reconstructive surgery therefore plays a vital role in achieving the best possible outcomes in the interdisciplinary treatment of soft-tissue sarcomas.

**Abstract:**

Although the involvement of plastic surgery has been deemed important in the treatment of sarcoma patients to avoid oncological compromises and ameliorate patient outcomes, it is not ubiquitously available. The accessibility of defect reconstruction and its therapeutic impact on sarcoma care is the subject of this analysis. Cross-sectional data from 1309 sarcoma patients were collected electronically at 39 German study centers from 2017 to 2019. A total of 621 patients with surgical treatment for non-visceral soft-tissue sarcomas were included. The associated factors were analyzed exploratively using multifactorial logistic regression to identify independent predictors of successful defect reconstruction, as well Chi-squared and Cochran–Mantel–Haenszel tests to evaluate subgroups, including limb-salvage rates in extremity cases. A total of 76 patients received reconstructive surgery, including 52 local/pedicled versus 24 free flaps. Sarcomas with positive margins upon first resection (OR = 2.3, 95%CI = 1.2–4.4) that were excised at centers with lower degrees of specialization (OR = 2.2, 95%CI = 1.2–4.2) were independently associated with the need for post-oncological defect coverage. In this context, the inhouse availability of plastic surgery (OR = 3.0, 95%CI = 1.6–5.5) was the strongest independent predictor for successful flap-based reconstruction, which in turn was associated with significantly higher limb-salvage rates (OR = 1.4, 95%CI = 1.0–2.1) in cases of extremity sarcomas (*n* = 366, 59%). In conclusion, consistent referral to specialized interdisciplinary sarcoma centers significantly ameliorates patient outcomes by achieving higher rates of complete resections and offering unrestricted access to plastic surgery. The latter in particular proved indispensable for limb salvage through flap-based defect reconstruction after sarcoma resection. In fact, although there remains a scarcity of readily available reconstructive surgery services within the current sarcoma treatment system in Germany, plastic and reconstructive flap transfer was associated with significantly increased limb-salvage rates in our cohort.

## 1. Introduction

Sarcomas, derived from the Ancient Greek words *σάρξ* (“sárx”, flesh) and *ωμα* (“ōma”, process), comprise a heterogenous group of malignant mesenchymal neoplasms, with the majority arising in soft tissues at potentially any anatomic region [1,2]. With an approximate annual incidence of 5 cases per 100,000 in Europe, adult soft-tissue (STSs) sarcomas are very rare tumors, most commonly affecting the extremities [3,4]. The most frequent STS subtypes include liposarcomas (LPSs), leiomyosarcomas (LMSs), and sarcoma not otherwise specified (NOS), with annual incidences of <1 per 100,000, respectively [4,5]. The relative frequency of each subtype varies, but the overall incidence of STSs typically increases with age towards a median peak around 66 years, generally afflicting both sexes equally [1,4,5]. The lower limb, particularly the thigh, is the most common site of origin, followed by the upper limb and trunk [6,7]. In sano resection continues to be the mainstay of curative treatment, aiming at improving the overall survival and locoregional recurrence rates [8,9]. Historically, amputation was regarded as the procedure of choice in the extremities [10,11,12,13]. Yet, numerous clinical studies have provided sufficient evidence that limb-sparing resections with wide surgical margins, combined with perioperative irradiation in more aggressive tumors, yield non-inferior survival rates [14,15,16,17,18]. Trunk-wall sarcomas are mostly located on the thoracic wall, where rapid progression along the costal periosteum and parietal pleura lead to the early involvement of critical structures and typically mandate full-thickness excisions that disintegrate skeletal stability [19,20].

Consequently, multimodal limb-sparing and function-preserving surgery in combination with adjuvant treatment regimens has now become the standard of care, thus considerably ameliorating functional outcomes without oncological compromises [21,22]. In this context, the involvement of reconstructive plastic surgery is recommended, as it can serve multiple purposes through the introduction of well-vascularized tissue to the site of tumor resection [23,24,25]. By making post-ablative reconstruction readily available, it can aid the oncologic surgeon in achieving wider and safer margins [26,27,28]. Thus, it can help extend the indications for limb salvage, allowing for an alternative to amputation [28,29,30,31]. By reducing and counteracting postoperative wound-healing complications, it allows for timely rehabilitation without delaying important adjuvant therapy [32,33,34,35,36]. By covering autologous transplants or synthetic semirigid and rigid implants for skeletal support in functional thoracic reconstruction, it facilitates extensive chest-wall resections [37,38]. By way of background, it is mandatory that the planning of both the sarcoma resection and subsequent defect reconstruction are agreed upon prior to the start of surgical treatment, according to the German S3-Sarcoma Guidelines [39]. In line with this, German Sarcoma Center accreditation requires that a dedicated plastic and reconstructive surgery unit be part of the treatment facilities or at least be contracted in written form.

However, in analogy to the limited experience of primary and secondary healthcare providers with STSs, which can prolong diagnostic intervals and delay specialist referrals, plastic and reconstructive surgery services are not ubiquitously available [40,41,42]. The aim of this present analysis was therefore to evaluate the state of accessibility of microvascular reconstructive surgery as well as its role and potential impact on the quality of medical care amongst the participants of the prospective nationwide PROSa (patient-reported outcome measures in sarcoma patients treated between 1984 and 2019) cohort study, which was conducted between September 2017 and February 2019 in 39 German study centers (ClinicalTrials.gov Identifier: NCT03521531), the data from which have previously been published elsewhere [43,44,45].

## 2. Materials and Methods

### 2.1. Study Population

Following approval by the local ethics committees of the Technical University of Dresden (EK1790422017) and all participating centers, cross-sectional data from 1309 adult patients with histological proof of sarcoma and their respective treating study centers were collected electronically by means of web-based questionnaires (REDCap, Vanderbilt University, Nashville, TN, USA). Minors, persons cognitively, mentally, or linguistically unable to complete the questionnaires, and patients unwilling to participate were excluded. In this present analysis, only patients with any form of surgical treatment of STSs not originating in visceral organs (i.e., intrathoracic and intraabdominal) or bone were assessed.

### 2.2. Explored Variables

Patient age and gender as well as present comorbidities were recorded at baseline (i.e., first in-center assessment at a PROSa study site after successful recruitment, regardless of current treatment status) along with all relevant histological, pathological, clinical, and anatomical details, including time since diagnosis, tumor type and subtype, detailed location, grading (G-status), size (T-status), and systemic spread (M-status). The present status (in treatment versus not in treatment) and intention of treatment (curative versus palliative) as well as the disease status at baseline (complete remission, partial remission, stable disease, or progression) were registered, along with any history of past treatments (surgery, neoadjuvant, and/or adjuvant therapy).

All cases of defect reconstruction by means of vascularized tissue according to plastic surgery principles were assessed in detail, including the type of reconstruction used (local and pedicled flaps (LPFs) versus free flaps (FFs)) as well as the timing of reconstruction (primary reconstruction, secondary reconstruction, as part of a resection of recurrent disease). In addition, the respective study centers were questioned as to whether a dedicated plastic surgery service was readily available at their institution or whether defect reconstruction required patient transfers elsewhere. In the latter case, the respective study centers were further questioned as to how and why the accessibility to plastic surgery defect reconstruction was restricted in their case and where they would send patients.

### 2.3. Statistical Analysis

Exploratory data analysis was performed with IBM SPSS Version 20 (IBM Corporation, Armonk, NY, USA) and RStudio Version 4.0.3 (RStudio, Inc., Boston, MA, USA) using the ‘stats’ package (version 4.2.0), the ‘compareGroups’ package (version 4.5.1), and the ‘epitools’ package (version 0.5–10.1). Data are presented as means with standard deviations (SDs) for continuous, medians with interquartile ranges (IQRs) for ordinal, and proportions with percentages (%) for categorical data. Pearson’s Chi-squared test and respective odds ratios with their corresponding 95% confidence intervals were used to assess group differences regarding categorical factors for 2 × 2 and 3 × 2 comparisons, whereas differences in the distribution of continuous variables were assessed using the unpaired Welch-corrected T-test or Mann–Whitney U-test. Differently distributed categorical and continuous data were entered into a multifactorial backward stepwise logistic regression model to identify predictive characteristics independently associated with the incidence of defect reconstruction via vascularized tissue transfer. A two-sided error probability of *p* < 0.05 was considered statistically significant.

## 3. Results

### 3.1. Patient and Tumor Characteristics

Of the entirety of the 39 study centers, 8 were private practices, 22 were maximum-care and university hospitals, and 9 were less specialized community hospitals or others. A total of 1309 patients consented to participate, 1139 of whom (87%) had at least had one therapeutic surgery. Cases of skeletal bone and chondrosarcomas, patients with sarcomas originating in visceral organs, and patients with gastrointestinal sarcomas were then excluded, constituting an analysis population of 621 cases of STS (see Figure 1). The predominant sarcoma subtypes were liposarcomas (*n* = 163, 26.2%), unclassified sarcomas (*n* = 145, 23.3%), and fibroblastic sarcomas (*n* = 123, 19.8%, see Table 1). The extremities were affected in the majority of cases (*n* = 366, 58.9%). Most tumors (*n* = 377, 75.9%) were classified as deep (i.e., subfascial). A total of 55.6% of patients were male (345) and 44.4% were female (276). The average age at diagnosis was 54.0 ± 15.1 years and the average age at baseline was 58.2 ± 14.8 years. Unlike the time of recruitment, the respective year of treatment of the individual patients included in the analysis ranged between 1984 and 2019.

### 3.2. Treatment Details and Oncological Outcomes

A total of 409 patients reported to have completed their treatment (65.9%) at baseline, whereas 145 were in the midst of undergoing therapy at the time of recruitment for the PROSa study (23.3%). The underlying treatment intention was curative in 78.3% of cases (*n* = 486) versus palliative in 20.5% (*n* = 127), and the majority of patients reported to be in remission (*n* = 335, 53.9%) or stable (*n* = 147, 23.7%) at baseline. Of the entirety of the 621 cases of sarcoma resections, 136 initial excisions (21.9%) were unplanned, and 293 primary surgeries (47.2%) did not result in negative margins. A total of 302 (48.6%) primary resections were carried out at university hospitals versus 213 (34.3%) at community hospitals. In 320 cases, locoregional radiotherapy was applied (51.5%), and in 191 cases (30.8%), systemic chemotherapy was administered. The in-study mortality rate throughout the course of the prospective observational period was 10.6%, and the local and systemic recurrence rates were 29.8% and 27.9%, respectively.

In 374 cases (60.2%), inhouse plastic surgery services were readily available. Amongst the remaining 247 cases (39.8%) with unavailable reconstructive surgery, the respective study centers reported unrestricted access to external plastic surgery services requiring patient transfer in 204 cases (82.6%) but acknowledged general barriers to reconstructive treatment options in as much as 17.4%. A total of 76 patients (12.2%) received reconstructive surgery for defect reconstruction as part of a multimodal treatment concept, including LPFs in 52 cases and FFs in 24 cases. Table 2 lists the respective flaps used in further detail. The most frequently used were local muscle and skin as well as the free Latissimus dorsi muscle flaps, respectively. A total of 316 of 366 limbs were salvaged (86.3%). 

### 3.3. Univariate Comparisons Regarding Defect Reconstruction

Table 3 and Table 4 list the unadjusted odds ratios (UORs) and their respective confidence intervals (95%CIs) of all the items assessed for group differences. Upon the comparison of cases with primary closure (*n* = 545, 87.8%) versus patients requiring vascularized tissue transfer (*n* = 76, 12.2%), higher-graded (G2–3: UOR = 1.96, 95%CI = 0.95–4.62) and smaller-sized (T1: UOR = 2.03, 95%CI = 1.10–3.67) tumors of the extremities (UOR = 1.93, 95%CI = 1.11–3.53) were significantly associated with flap coverage. Regarding the details of the primary resection, unplanned (UOR = 2.06, 95%CI = 1.21–3.44) and incomplete excisions (R1: UOR = 2.25, 95%CI = 1.27–3.03) that were carried out at centers with lower levels of expertise (community hospitals: UOR = 1.79, 95%CI = 1.04–3.09) were at a higher risk of necessitating flap-based reconstruction. The overall treatment intention at baseline was almost three times more likely to be curative amongst the reconstructed cohort (UOR = 2.78, 95%CI = 1.33–6.85), with flap-covered patients exhibiting an over-three-times-higher chance of remission at baseline (UOR = 3.52, 95%CI = 1.65–8.76). The overall limb-salvage rate in cases of extremity sarcomas (*n* = 366, 58.9%) was significantly higher after flap transfer as opposed to primary closure (49 of 53 limbs = 92.5% versus 267 of 313 limbs = 85.3%, UOR = 2.87, 95%CI = 1.32–7.57), with no significant differences in local recurrence and survival. Finally, the inhouse availability of a dedicated plastic surgery service increased the likelihood of receiving defect reconstruction by about three-fold (available: UOR = 2.74, 95%CI = 1.57–5.04).

### 3.4. Multifactorial Logistic Regression Regarding Defect Reconstruction

The covariates entered into the backward stepwise multifactorial binary regression model were: tumor grading, tumor size, tumor location, tumor depth, initial treatment intention, nature (planned versus unplanned), outcome (R0, R1, R2, and Rx margins), facility type (university hospital, hospital of maximum care, community hospital, private practice) of the primary resection, limb-salvage rates, and inhouse availability of plastic and reconstructive surgery. The final model included tumor grading, size and depth, treatment intention, resection margins, limb-salvage rates, the availability of plastic surgery, and the type of resecting center. Table 5 lists the resulting adjusted odds ratios (AORs) and corresponding confidence intervals (95%CIs) for all independent predictors of flap-based reconstruction. Accordingly, positive margins (R1: AOR = 2.28, 95%CI = 1.17–4.58, *p* = 0.015) and having the first resection at centers of lower expertise (community hospitals: AOR = 2.24, 95%CI = 1.19–4.23, *p* = 0.015) were independently associated with tissue-transfer-based defect reconstruction, particularly in cases with readily available plastic surgery services (AOR = 2.95, 95%CI = 1.58–5.50, *p* < 0.001).

### 3.5. Exploratory Subgroup Analyses

The risk of extremity amputation was significantly higher for patients treated at centers without readily available plastic surgery services (pooled OR = 1.43, 95%CI = 1.03–2.06, *p* = 0.035, see Figure 2, Panel A). This bivariate association was further impacted by the type of reconstruction used, with the highest relative salvage rates after free flaps and the lowest after primary closure. Furthermore, tumor grading (with the highest risk of amputation in high-grade sarcomas), tumor size (with the highest risk of amputation in larger T2–T4 sarcomas), tumor depth (with the highest risk of amputation in deep sarcomas), and positive margins upon primary excision (with the highest risk of amputation in macroscopically incomplete resections) also influenced the limb-salvage rates. Figure 2 visualizes the subgroup differences with an emphasis on the impact of plastic surgery availability.

## 4. Discussion

### 4.1. Main Findings Put in Context

In this large prospective multi-center observational study, we were able to show that the availability of reconstructive surgery in the form of an inhouse plastic surgery service was a statistically significant independent predictor of defect reconstruction following sarcoma resection, particularly in high-grade tumors and cases of incomplete excisions with positive margins. Accordingly, readily available plastic surgery and flap-based reconstructions were associated with significantly higher chances of limb salvage without any compromise in overall survival or local recurrence rates, even despite more aggressive tumors and higher rates of positive margins. Of further note, amongst those patients with plastic surgery defect reconstructions, recipients of microsurgical free flaps achieved the highest overall limb-salvage rates.

While the first flap-based chest-wall reconstruction was described in 1906, the underlying concept of transferring well-vascularized tissue to the site of sarcoma resection defects in order to avoid amputations and enable limb-sparing surgery was first described in 1986 by Usui and co-workers [29]. In this context, our findings of independently increased limb-salvage rates in patients with defect reconstruction according to plastic surgery principles generally fit in well with the previously published reports [30,32]. In addition, recent works by Slump, Götzl, and Dadras et al. have repeatedly demonstrated that flap-based reconstructive approaches do not increase local recurrence or overall mortality rates, which is well in line with our findings [46,47,48]. Yet, by evaluating the admission details of STS patients to their department in Spain between 2000 and 2010, Marré et al. found that about half of all cases were late referrals with higher subsequent complication rates, leading them to both emphasize the importance of early plastic surgery participation as well as lament the lack of clear-cut referral guidelines [42]. They concluded that a considerable number of STS patients received insufficient reconstructions due to the limited availability of reconstructive surgery, thus worsening surgical and functional outcomes [42]. In fact, Agrawal et al. further elaborated on the practical implications of this important issue when they showed that amputation rates dropped significantly following the creation of a dedicated plastic surgery service at Ohio State University in 2007 [28].

It is therefore all the more surprising that approximately one-third of all participating centers in the PROSa study indicated that inhouse plastic surgery services were not available to them. Amongst those centers with no inhouse plastic surgery service, one-fifth even pointed out veritable barriers to reconstructive surgery that they would regularly experience when trying to transfer their patients. Our study therefore directs attention towards this thus far underreported problem for the first time. Indeed, for sarcoma patients, there seems not only to be a long diagnostic interval, but there is now empirical evidence of a considerable gap in plastic surgery availability.

### 4.2. Further Results Put in Context

We found a marked heterogeneity of sarcoma subtypes amongst all the included patients, which is in accordance with the previous publications. Liposarcomas, which were found in one-fourth of the cases in our cohort, also constituted the most common tumor subtype in Dadras’s study population with a 25% incidence rate [48,49]. The distribution of other tumor characteristics, such as location, depth, size, and grading, albeit being only inconsistently reported, were largely comparable to the current literature [46,47,48]. Particularly, as previously observed by Dadras and Slump, a smaller size and superficial depth, while seemingly counterintuitive, were associated with an increased likelihood of post-ablative flap coverage in our patients [46,48]. This may be due to larger and deeper tumors resulting in greater ablation cavities facilitating primary closure, whereas smaller superficial tumors often require the resection of adjacent skin and subcutaneous tissue. This notion is further supported by the increased odds of requiring flap coverage in the extremities, where both the soft tissue and skin envelope offer little excess for subsequent primary closure. In line with this, the need for flap coverage amongst our cohort was highest in the distal extremities (hand and wrist: *n* = 2/7, 29%; foot and ankle: *n* = 6/20, 30%). Although extremity sarcomas represented the largest proportion of the cases in our study (*n* = 366, 59%), a considerable percentage of tumors was located on the trunk (*n* = 212, 34%). Interestingly, amongst these, the number of defects requiring flap-based reconstructions was unevenly distributed: a disproportionately greater number of chest-wall (*n* = 9/63, 14%) and groin-girdle defects (*n* = 7/77, 9%) received flap coverage. As opposed to the abdominal wall, particularly the anterolateral and upper chest wall, as well as the groin, have been described as being prone to mandating tissue transfer, due to a marked superficiality of vital structures paired with a sparsity of locally available pliable tissues [50,51,52]. Regarding the choice of flaps, there was a stark heterogeneity, probably due to the variety of contributing study centers, which encompassed highly sophisticated microsurgery units as well as community hospitals of standard care and private practices. Yet, the observed 2:1 ratio in favor of local versus free flaps is analogous to data sets from maximum-care academic hospitals with specialized plastic surgery departments [46,48]. Furthermore, the 52% locoregional radiotherapy and 31% systemic chemotherapy rates we observed in our population also correspond well to the literature data [53].

What stands out, however, is the combination of high incidences of inadvertent first resections (reaching over 20% in our cohort) and non-negative initial margins (amounting to over 47% in our cohort), which are findings that have been shown to be mutually dependent. In fact, unplanned excisions have been linked to increased rates of positive margins and local recurrences, requiring wider re-excisions and necessitating subsequent flap coverage [54,55]. In this context, tumor ablation at academic and specialized centers has been shown to result in higher rates of negative margins in a recent analysis of referral pathways, highlighting differences in the geographic access to proper sarcoma care in the United States [56]. In fact, the Scandinavian Sarcoma Register Group has repeatedly demonstrated that the implementation of specialized sarcoma centers can help improve surgical margins and local recurrence rates [57,58]. In line with these observations, Götzl and co-workers noted that an alarming 91% of all patients that had initially been treated at external and presumably less specialized institutions were transferred to their department with non-negative margins [47]. Our findings of an initial R1 status independently increasing the need for subsequent flap-based defect reconstruction further emphasize the implication of inadvertent and incomplete sarcoma excisions for plastic surgeons.

### 4.3. Strengths and Limitations

The PROSa study is one of the largest prospective observational studies in sarcoma patients published thus far [43,44,45]. Cases from 39 study centers encompassing hospitals of maximum care as well as community hospitals and private practices were included. The aggregate of participating study centers therefore reflects the current state of sarcoma care in Germany and allows important inferences to be drawn on the underlying referral and treatment networks. In this regard, previously published reports, which were mostly based on monocentric studies with narrow inclusion criteria, collectively failed to adequately depict the unbiased therapeutic reality and were thus hindered by an inherently limited generalizability. The PROSa study, on the contrary, thoroughly depicts the current situation of sarcoma patients and care providers in Germany, thus shedding light on the big picture.

Yet, this study, too, is not without limitations. First and foremost, as is the case with any observational study, causal relationships cannot be deduced. Secondly, our study may be subject to an inherent selection bias both on the subordinate level of the respective study centers as well as on the individual patient level. Regarding the former issue, it must be acknowledged that the majority of included patients was recruited at university hospitals or specialized centers. Despite its broad approach and comprehensive nature, the PROSa study might therefore still not be fully representative of the collectivity of sarcoma patients in Germany. Regarding the latter aspect, further selection bias is also probable in the form of possible sick survivor bias, as healthy individuals are generally less likely to visit recruiting study centers. Third, due to the cross-sectional study design, the main cohorts of primary closure versus defect reconstruction are not only ill-balanced, but also suggestive of ill-controlled confounders. This becomes evident in view of a significant heterogeneity in treatment intention and disease status between both groups, with significantly more curative cases and remissions amongst the reconstructed patients, for example. Furthermore, due to the low number of reconstructed patients, further subgroup analyses were hindered by a considerable sparsity of data. Therefore, all stratified subgroup analyses were underpowered, failing to reach statistical significance, and were carried out exploratorily only. Fourth, it has to be acknowledged that the majority of our cross-sectional cohort had already completed their treatment at baseline, thus impeding prospective evaluations of the impact of plastic surgery in the therapeutic pathway of sarcoma patients and survivors. In fact, approximately 2% of patients had received their first surgical treatment before the year 2000 and 14% before the year 2010. Finally, surgical outcomes of potential further interest other than resection margins and amputation rates, such as complications necessitating revisional surgeries, most importantly flap losses, were not assessed.

## 5. Conclusions

In this large prospective multi-center study, we were able to show that non-negative soft-tissue sarcoma resections carried out at centers with lower degrees of specialization significantly increased the need for subsequent flap-based defect coverage. In this context, unrestricted access to dedicated plastic surgery services at specialized interdisciplinary centers proved to be of paramount importance to achieve limb salvage through successful defect reconstruction. Nevertheless, our study provides evidence that there still remains a considerable scarcity of readily available reconstructive surgery resources within the current multimodal sarcoma treatment system in Germany.

## Figures and Tables

**Figure 1 cancers-14-04312-f001:**
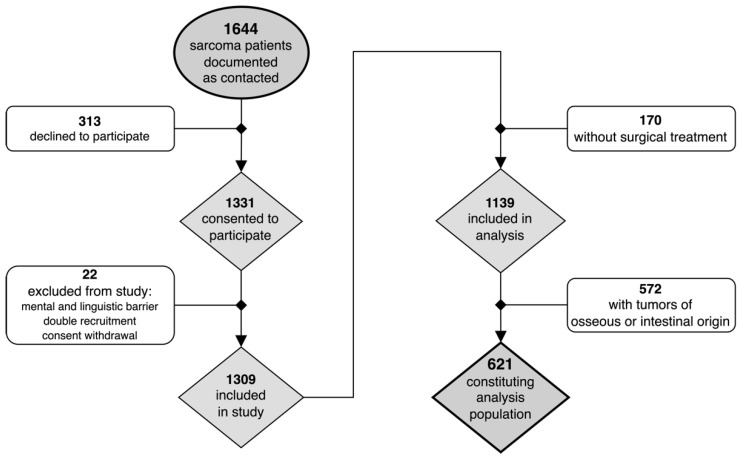
Flow diagram of patients recruited as part of the original PROSa study protocol and eligibility criteria for this present analysis.

**Figure 2 cancers-14-04312-f002:**
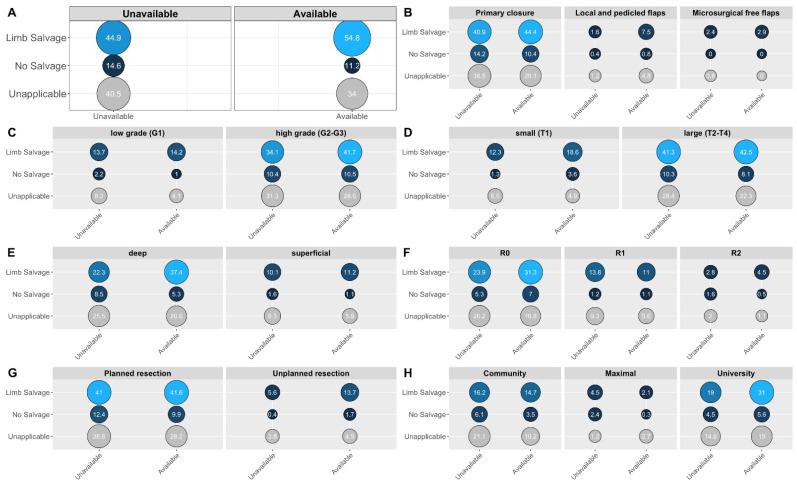
Balloon plots of exploratory subgroup analyses illustrating higher limb-salvage rates (rows, bottom row level “unapplicable” represents non-extremity sarcomas) with availability of plastic and reconstructive surgery services (columns). Accordingly, centers with readily available plastic surgeons were associated with a significantly higher overall chance of limb salvage (pooled OR = 1.43, 95%CI = 1.03–2.06, *p* = 0.035) (**A**). Further surgical subgroup differences impacting this association included: type of defect closure (**B**), tumor grading (unequivocally classified cases not shown) (**C**), tumor size (unequivocally classified cases not shown) (**D**), tumor depth (unequivocally classified cases not shown) (**E**), and margins of first resection (cases of unknown or unclear margins not shown) (**F**). Numbers represent column-wise cumulative percentages.

**Table 1 cancers-14-04312-t001:** Sarcoma subtypes.

Sarcoma Subtypes	Frequency	Percentage
Liposarcoma	*n* = 163	26.2%
Fibroblastic, myofibroblastic, fibrohistiocytic sarcoma	*n* = 123	19.8%
Leiomyosarcoma	*n* = 72	11.6%
Synovial sarcoma	*n* = 40	6.4%
Vascular tumors	*n* = 15	2.4%
Nerve-sheath tumors	*n* = 15	2.4%
Tumors of uncertain differentiation	*n* = 15	2.4%
Skeletal muscle tumors	*n* = 11	1.8%
Extraosseous chondrosarcoma	*n* = 11	1.8%
Extraosseous ewing sarcoma	*n* = 6	1.0%
Extraosseous osteosarcoma	*n* = 4	0.6%
Stroma sarcoma	*n* = 1	0.2%
Unclassified sarcoma	*n* = 145	23.3%

The most frequent subtypes were liposarcomas (26%), followed by unclassified sarcomas (23%), and fibroblastic sarcomas (20%).

**Table 2 cancers-14-04312-t002:** Flap types.

Flap Types	Frequency	Percentage
Local and pedicled flaps (LPFs):	*n* = 52	68.4%
Local muscle flap	*n* = 21	40.4%
Local skin flap	*n* = 19	36.5%
Pedicled Latissimus dorsi muscle flap	*n* = 4	7.7%
Pedicled ALT flap	*n* = 2	3.8%
Pedicled Sartorius muscle flap	*n* = 2	3.8%
Pedicled Gluteus muscle flap	*n* = 2	3.8%
Pedicled Pectoralis muscle flap	*n* = 1	1.9%
Pedicled freestyle perforator flap	*n* = 1	1.9%
Free microsurgical flaps (FFs):	*n* = 24	31.6%
Free Latissimus dorsi muscle flap	*n* = 9	37.5%
Free ALT flap	*n* = 6	25.0%
Free PSC flap	*n* = 5	20.8%
Free VRAM muscle flap	*n* = 1	4.2%
Free TAP flap	*n* = 1	4.2%
Free TFL muscle flap	*n* = 1	4.2%
Free upper arm flap	*n* = 1	4.2%

A total of 76 patients received vascularized tissue transfers according to plastic surgery principles, including 52 local and pedicled (LPFs) versus 24 free flaps (FFs). The most frequently transferred tissues were local muscle (28%) and skin (25%), as well as the free Latissimus dorsi muscle (12%) and ALT perforator (8%) flaps, respectively (ALT = anterior lateral thigh, PSC = parascapular, TAP = thoracodorsal artery perforator, TFL = tensor fasciae latae, VRAM = vertical rectus abdominis myocutaneous).

**Table 3 cancers-14-04312-t003:** Description of study population.

	Entire Cohort	Defect Reconstruction	Primary Closure	Univariate Odds Ratios	Univariate PError Robability
	621 (100%)	76 (12.2%)	545 (87.8%)		
**Tumor characteristics**					
Tumor grading (*n* = 477 ^§^):					0.077
High grade (G2–3)	376 (78.8%)	55 (87.3%)	321 (77.5%)	Reference	
Low grade (G1)	101 (21.2%)	8 (12.7%)	93 (22.5%)	0.51 (0.22; 1.06)	
Unclassified ^§^	144 ^§^ (23.2%)	13 (17.1%)	131 (24.0%)		
Tumor size (*n* = 402 ^$^):					0.028
Large (T2–4)	304 (75.6%)	36 (63.2%)	268 (77.7%)	Reference	
Small (T1)	98 (24.4%)	21 (36.8%)	77 (22.3%)	2.03 (1.10; 3.67)	
Unclassified ^$^	219 ^$^ (35.3%)	19 (25.0%)	200 (36.7%)		
Tumor location (*n* = 621):					0.069
Trunk	212 (34.1%)	17 (22.4%)	195 (35.8%)	Reference	
Pelvis and groin	77 (12.4%)	7 (9.2%)	70 (12.8%)		
Abdominal wall, lower back, retroperitoneum	72 (11.6%)	1 (1.3%)	71 (13.0%)		
Thoracic wall and upper back	63 (10.1%)	9 (11.8%)	54 (9.9%)		
Extremities	366 (58.9%)	53 (69.7%)	313 (57.4%)	1.93 (1.11; 3.53)	
Lower Extremity	308 (49.8%)	44 (57.9%)	264 (48.6%)	1.76 (1.02; 3.14)	
Thigh	233 (37.6%)	24 (31.6%)	209 (38.5%)		
Lower leg	55 (8.9%)	14 (18.4%)	41 (7.6%)		
Foot and ankle	20 (3.2%)	6 (7.9%)	14 (2.6%)		
Upper Extremity	58 (9.4%)	9 (11.8%)	49 (9.0%)	1.96 (0.80; 4.48)	
Upper arm	33 (5.3%)	6 (7.9%)	27 (5.0%)		
Forearm	18 (2.9%)	1 (1.3%)	17 (3.1%)		
Hand	7 (1.1%)	2 (2.6%)	5 (0.9%)		
Other incl. head and neck	43 (6.9%)	6 (7.9%)	37 (6.8%)	1.88 (0.63; 4.92)	
Tumor depth (*n* = 497 ^†^):					0.177
Deep	377 (75.9%)	43 (68.3%)	334 (77.0%)	Reference	
Superficial	120 (24.1%)	20 (31.7%)	100 (23.0%)	1.56 (0.86; 2.74)	
Unclassified ^†^	124 ^†^ (20.0%)	13 (17.1%)	111 (20.4%)		
**Patient characteristics**					
Gender (*n* = 621):					0.575
Female	276 (44.4%)	31 (40.8%)	245 (45.0%)	Reference	
Male	345 (55.6%)	45 (59.2%)	300 (55.0%)	1.18 (0.73; 1.94)	
Age at diagnosis (years) (*n* = 621):	54.0 (15.1)	57.5 (14.6)	53.5 (15.2)	1.02 (1.00; 1.04)	0.027
Age at baseline (years) (*n* = 621):	58.2 (14.8)	62.5 (14.8)	57.6 (14.7)	1.02 (1.01; 1.04)	0.008
Year of first resection (scaled) (*n* = 621):	0.00 (4.87)	0.11 (4.61)	−0.78 (6.41)	0.97 (0.93; 1.01)	0.246
Disease status at baseline (*n* = 621):					0.001
In remission	335 (53.9%)	51 (67.1%)	284 (52.1%)	Reference	
Stable disease	147 (23.7%)	7 (9.2%)	140 (25.7%)	0.28 (0.11; 0.61)	
Progression	95 (15.3%)	8 (10.5%)	87 (16.0%)	0.52 (0.22; 1.09)	
Unclear	44 (7.1%)	10 (13.2%)	34 (6.2%)		
**Treatment characteristics**					
Treatment intention at baseline (*n* = 621):					0.013
Curative	486 (78.3%)	69 (90.8%)	417 (76.5%)	Reference	
Palliative	127 (20.5%)	7 (9.2%)	120 (22.0%)	0.36 (0.15; 0.75)	
Unclear	8 (1.3%)	0 (0.00%)	8 (1.5%)		
Treatment status at baseline (*n* = 621):					0.503
Completed	409 (65.9%)	54 (71.1%)	355 (65.1%)	Reference	
Active	145 (23.3%)	18 (23.7%)	127 (23.3%)	0.94 (0.52; 1.63)	
Planned	44 (7.1%)	3 (4.0%)	41 (7.5%)	0.50 (0.11; 1.46)	
Paused	23 (3.7%)	1 (1.3%)	22 (4.0%)	0.34 (0.01; 1.67)	
Type of first treatment (*n* = 621):					0.882
Surgery	492 (79.2%)	63 (82.9%)	429 (78.7%)	Reference	
Chemotherapy	79 (12.7%)	8 (10.5%)	71 (13.0%)	0.78 (0.33; 1.61)	
Radiotherapy	42 (6.8%)	4 (5.3%)	38 (7.0%)	0.74 (0.21; 1.93)	
Unknown or unclear	8 (1.3%)	1 (1.3%)	7 (1.3%)		

^§^ In 144 cases (23.2%), tumor grade (G-status) was not available. ^$^ In 219 cases (35.3%), tumor size (T-status) was not available. ^†^ In 124 cases (20.0%), tumor depth was not available. A total of 621 patients were included in this present analysis, aged 58 years on average, with an equal gender distribution. Most patients suffered from larger-sized and higher-graded deep STSs located on the extremities. At baseline, the majority of treatment intentions were curative and the majority of patients had already completed their treatment.

**Table 4 cancers-14-04312-t004:** Description of treatment details.

	Entire Cohort	Defect Reconstruction	Primary Closure	Univariate Odds Ratios	Univariate Error Probability
	621 (100%)	76 (12.2%)	545 (87.8%)		
**Details of surgical treatment**					
Facility type upon first resection (*n* = 621):					0.056
University hospital	302 (48.6%)	28 (36.8%)	274 (50.3%)	Reference	
Hospital of maximum care	39 (6.3%)	4 (5.3%)	35 (6.4%)	1.15 (0.32; 3.18)	
Community hospital	213 (34.3%)	33 (43.4%)	180 (33.0%)	1.79 (1.04; 3.09)	
Private practice	33 (5.3%)	8 (10.5%)	25 (4.6%)	3.15 (1.22; 7.46)	
Unknown or unclear	34 (5.5%)	3 (4.0%)	31 (5.7%)		
Nature of first resection (*n* = 621):					0.009
Planned	485 (78.1%)	50 (65.8%)	435 (79.8%)	Reference	
Unplanned	136 (21.9%)	26 (34.2%)	110 (20.2%)	2.06 (1.21; 3.44)	
Margins achieved in first resection (*n* = 621):					0.041
R0	328 (52.8%)	34 (44.7%)	294 (53.9%)	Reference	
R1	126 (20.3%)	26 (34.2%)	100 (18.3%)	2.25 (1.27; 3.93)	
R2	39 (6.3%)	5 (6.6%)	34 (6.2%)	1.30 (0.42; 3.31)	
Rx	38 (6.1%)	3 (4.0%)	35 (6.4%)	0.77 (0.17; 2.32)	
Unknown or unclear	90 (14.5%)	8 (10.5%)	82 (15.0%)		
Limb-salvage rates (at t2) (*n* = 621):					0.047
LSS	316 (50.9%)	49 (64.5%)	267 (49.0%)	Reference	
Amputations	26 (4.2%)	2 (2.6%)	24 (4.4%)	0.49 (0.07; 1.72)	
Unapplicable	227 (36.6%)	23 (30.3%)	204 (37.4%)	0.62 (0.36; 1.04)	
Unknown or unclear	52 (8.4%)	2 (2.6%)	50 (9.2%)		
Availability of reconstructive surgery (*n* = 621):					0.001
Available	374 (60.2%)	60 (78.9%)	314 (57.6%)	Reference	
Unavailable	247 (39.8%)	16 (21.1%)	231 (42.4%)	0.37 (0.20; 0.64)	
**Details of adjuvant treatment**					
Radiotherapy (diagnosis–t2) (*n* = 621):					0.688
No radiotherapy	301 (48.5%)	36 (47.4%)	265 (48.6%)	Reference	
Neoadjuvant	79 (12.7%)	12 (15.8%)	67 (12.3%)	1.33 (0.63; 2.63)	
Adjuvant	241 (38.8%)	28 (36.8%)	213 (39.1%)	0.97 (0.57; 1.64)	
Chemotherapy (diagnosis–t2) (*n* = 621):					0.872
No chemotherapy	430 (69.2%)	53 (69.7%)	377 (69.2%)	Reference	
Neoadjuvant	101 (16.3%)	11 (14.5%)	90 (16.5%)	0.88 (0.42; 1.70)	
Adjuvant	90 (14.5%)	12 (15.8%)	78 (14.3%)	1.10 (0.54; 2.10)	
**Oncological Outcomes**					
Overall mortality (t0–t2) ^‡^ (*n* = 621):					0.155
Alive	555 (89.4%)	72 (94.7%)	483 (88.6%)	Reference	
Deceased	66 (10.6%)	4 (5.3%)	62 (11.4%)	0.45 (0.13; 1.13)	
Local recurrences (t0–t2) ^‡^ (*n* = 621):					0.517
None	433 (69.7%)	49 (64.5%)	384 (70.5%)	Reference	
Recurrences	185 (29.8%)	27 (35.5%)	158 (29.0%)	1.34 (0.80; 2.21)	
Unknown or unclear	3 (0.5%)	0 (0.00%)	3 (0.6%)		
Systemic lesions (t0–t2) ^‡^ (*n* = 621):					0.070
None	439 (70.7%)	60 (78.9%)	379 (69.5%)	Reference	
Metastases	173 (27.9%)	14 (18.4%)	159 (29.2%)	0.56 (0.29; 1.01)	
Unknown or unclear	9 (1.5%)	2 (2.6%)	7 (1.3%)		

^‡^ (t0–t2) = time interval between baseline (individual study inclusion) and end of study (death, loss to follow-up, end of observation). Primary surgeries comprised unplanned resections in 22% and resulted in non-negative margins in 47%. In approximately one-third of cases, inhouse plastic and reconstructive surgery services were not available. Amongst these, access to external plastic surgery services was reported as being restricted in as many as one-fifth of cases (LSS = limb-sparing surgery).

**Table 5 cancers-14-04312-t005:** Independent Predictors of Defect Reconstruction.

	Entire Cohort	Defect Reconstruction	Primary Closure	Adjusted Odds Ratios	Adjusted Error Probability
	621 (100%)	76 (12.2%)	545 (87.8%)		
**Tumor characteristics**					
Tumor grading:					
High grade (G2–3)	376 (78.8%)	55 (87.3%)	321 (77.5%)	1.98 (0.86; 4.52)	0.10
Tumor size:					
Small (T1)	98 (24.4%)	21 (36.8%)	77 (22.3%)	1.45 (0.75; 2. 97)	0.27
Tumor location:					
Extremities	366 (58.9%)	53 (69.7%)	313 (57.4%)	1.47 (0.81; 2.67)	0.20
**Treatment characteristics**					
Treatment intention at baseline:					
Curative	486 (78.3%)	69 (90.8%)	417 (76.5%)	1.94 (0.83; 4.58)	0.13
**Details of surgical treatment**					
Margins at first ablative surgery:					
R1	141 (16.9%)	27 (28.4%)	114 (15.5%)	2.28 (1.17; 4.44)	0.015
Availability of reconstructive surgery:					
Available	374 (60.2%)	60 (78.9%)	314 (57.6%)	2.95 (1.58; 5.50)	<0.001
Facility type upon first resection					
University hospital	302 (48.6%)	28 (36.8%)	274 (50.3%)	0.47 (0.24; 0.84)	0.015
Community hospital	213 (34.3%)	33 (43.4%)	180 (33.0%)	2.24 (1.19; 4.23)	0.015

Incomplete resections (AOR = 2.3) carried out at facilities with a lower level of expertise (AOR = 2.2) and readily available plastic and reconstructive surgery services (AOR = 3.0) were independently associated with patients receiving defect reconstructions, particularly in cases of smaller-sized and higher-graded tumors of the extremities.

## Data Availability

The data presented in this study are available on request from the corresponding authors.

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
