# Peer review of "The Therapeutic Role of Plastic and Reconstructive Surgery in the Interdisciplinary Treatment of Soft-Tissue Sarcomas in Germany—Cross-Sectional Results of a Prospective Nationwide Observational Study (PROSa)"

_cancers, 2022, doi:10.3390/cancers14174312_

Round 1
Reviewer 1 Report
Remarkable study with very interesting findings. Congratulations to the authors!
This paper definitely has to be published in a high impact factor journal.
It is well known that patients treated in sarcoma specialized centers do have better outcomes. It is also known that taking care of the patient in such specialized centers do yield better results in terms of R0 resections. Although logic, it is however less known that having a plastic surgeon readily available also contributes significantly to the outcome of the patient.
This is definitely yet another strong message to the surgical/medical community to send those patients to specialized centers without delays.
Furthermore, the authors demonstrate that the rate of reinterventions due to unplanned/incomplete surgeries rendered subsequent treatment more complicated and more frequently associated to the necessity of a flap.
I would greatly appreciate a stronger abstract and conclusions depicting those facts as such.
The message is strong but is not as striking in the abstract and the conclusion.
Details
44-46: “In summary, incomplete tumor resections carried out at centers with lower degrees of specialization on the one hand, and readily available plastic surgery on the other hand were independently associated with the implementation of defect reconstruction – which in turn was associated with higher chances of limb preservation.”
This phrase is a bit confusing: it reads as if incomplete tumor resection in centers with low degree of specialization leads to higher chances of limb sparing… I think it is not the intended message.
334-336: What is more, in almost 20% of those cases with no inhouse plastic surgery service, the respective treating study centers even pointed out veritable barriers to reconstructive surgery they would experience on a regular basis when trying to transfer their patients.
Phrase is a bit confusing as well. Maybe try and shorten it a bit.
Author Response
Reviewer 1
Remarkable study with very interesting findings. Congratulations to the authors!
This paper definitely has to be published in a high impact factor journal.
It is well known that patients treated in sarcoma specialized centers do have better outcomes. It is also known that taking care of the patient in such specialized centers do yield better results in terms of R0 resections. Although logic, it is however less known that having a plastic surgeon readily available also contributes significantly to the outcome of the patient. This is definitely yet another strong message to the surgical/medical community to send those patients to specialized centers without delays. Furthermore, the authors demonstrate that the rate of reinterventions due to unplanned/incomplete surgeries rendered subsequent treatment more complicated and more frequently associated to the necessity of a flap. I would greatly appreciate a stronger abstract and conclusions depicting those facts as such. The message is strong but is not as striking in the abstract and the conclusion.
We are delighted about the enthusiasm this reviewer is expressing after having read our manuscript!
Both the abstract and conclusion have been altered accordingly to emphasize not only the importance of treating sarcoma patients in specialized centers but also having plastic surgeons readily available.
Details
44-46: “In summary, incomplete tumor resections carried out at centers with lower degrees of specialization on the one hand, and readily available plastic surgery on the other hand were independently associated with the implementation of defect reconstruction – which in turn was associated with higher chances of limb preservation.”
This phrase is a bit confusing: it reads as if incomplete tumor resection in centers with low degree of specialization leads to higher chances of limb sparing... I think it is not the intended message.
334-336: What is more, in almost 20% of those cases with no inhouse plastic surgery service, the respective treating study centers even pointed out veritable barriers to reconstructive surgery they would experience on a regular basis when trying to transfer their patients.
Phrase is a bit confusing as well. Maybe try and shorten it a bit.
Both phrases have been altered according to this reviewer’s suggestion:
“In summary, unplanned and incomplete primary tumor resections carried out at centers with lower degrees of specialization significantly increased the subsequent need for flap-based defect coverage. In line with this, a readily available team of plastic surgeons was independently associated with successful defect reconstruction – which in turn was associated with significantly higher chances of limb preservation.”
“It is therefore all the more surprising that approximately one third of all participating centers in the PROSa study indicated that inhouse plastic surgery services were not available to them. Amongst those centers with no inhouse plastic surgery service, one fifth even pointed out veritable barriers to reconstructive surgery they would regularly experience when trying to transfer their patients.”
Reviewer 2 Report
This is a very interesting work facing the reality of sarcoma treatment in Germany. This scenario can be a representative example of Europe and western world.
The role of plastic surgery is essential in sarcoma surgery and centers treating sarcoma shall include a plastic surgery team. This is a fact that authors perfectly reflect along the paper.
However, I would like to suggest some corrections:
1. When reading this work I have some doubts about the role of outpatient offices. You need to explain it better. Are these ones centers associated to big hospitals or regional centers?.
2. About results, unclassified sarcomas are placed in second position. Please, let us know why you consider it into this classification.
3. About reconstructive procedures: You mention local muscle flaps, but then you mention some important muscle flaps like ALT, LD.... I need to know what you mean when writing local muscle flaps.
4. Skeleton and bone sarcomas, visceral have been excluded. You say it in results. This must be clarified previously in materials and method.
Author Response
Reviewer 2
This is a very interesting work facing the reality of sarcoma treatment in Germany. This scenario can be a representative example of Europe and western world.
The role of plastic surgery is essential in sarcoma surgery and centers treating sarcoma shall include a plastic surgery team. This is a fact that authors perfectly reflect along the paper.
However, I would like to suggest some corrections:
- When reading this work I have some doubts about the role of outpatient offices. You need to explain it better. Are these ones centers associated to big hospitals or regional centers?
This is an important question: rather than hospital-associated outpatient/ambulatory centers, those were (physician-owned) private practices. This has now been corrected accordingly.
- About results, unclassified sarcomas are placed in second position. Please, let us know why you consider it into this classification.
The order of sarcoma types was purely quantitative (in order of decreasing frequency). For the sake of clarity, however, unclassified sarcomas were now moved to the bottom.
- About reconstructive procedures: You mention local muscle flaps, but then you mention some important muscle flaps like ALT, LD.... I need to know what you mean when writing local muscle flaps.
For the sake of conciseness, we combined both local and pedicled flaps into one group. In this context, all local tissue flaps that did not require microsurgical dissection of the pedicle (i.e. advancement of a local gastrocnemius or peroneus brevis muscle flap) were classified as “local” flaps. On the contrary, pedicled flaps requiring extensive microsurgical dissection of the pedicle (i.e., pedicled ALT perforator or Latissimus dorsi muscle flaps) were classified as “pedicled” flaps and those were subsequently listed separately.
- Skeleton and bone sarcomas, visceral have been excluded. You say it in results. This must be clarified previously in materials and method.
This has now been clarified in the Materials and Methods section (2.1. Study Population):
“[…] In this present analysis, only patients with any form of surgical treatment of STS not originating in visceral organs (i.e., intrathoracal and intraabdominal) or bone were assessed. […]”